Trait–fitness associations do not predict
within-species phenotypic evolution
over 2 million years. *Proc. R. Soc. B* **288**:
20202047.

palaeontology, evolution, ecology

phenotypic selection, fossil time series,
fitness component, palaeoclimate,
ecological interactions, Pleistocene

**Authors for correspondence:**
Emanuela Di Martino
e-mail: e.d.martino@nhm.uio.no
Lee Hsiang Liow
e-mail: l.h.liow@nhm.uio.no

Electronic supplementary material is available
online at https://doi.org/10.6084/m9.figshare.
c.5269037.

# Trait–fitness associations do not predict within-species phenotypic evolution over 2 million years

Emanuela Di Martino[1] and Lee Hsiang Liow[1,2]

[1]Natural History Museum, and [2]Centre for Ecological and Evolutionary Synthesis, Department of Biosciences, University of Oslo, Oslo, Norway

EDM, 0000-0002-3892-4036; LHL, 0000-0002-3732-6069

Long-term patterns of phenotypic change are the cumulative results of tens of thousands to millions of years of evolution. Yet, empirical and theoretical studies of phenotypic selection are largely based on contemporary populations. The challenges in studying phenotypic evolution, in particular trait–fitness associations in the deep past, are barriers to linking micro- and macroevolution. Here, we capitalize on the unique opportunity offered by a marine colonial organism commonly preserved in the fossil record to investigate trait–fitness associations over 2 Myr. We use the density of female polymorphs in colonies of *Antartothoa tongima* as a proxy for fecundity, a fitness component, and investigate multivariate signals of trait–fitness associations in six time intervals on the backdrop of Pleistocene climatic shifts. We detect negative trait–fitness associations for feeding polymorph (autozooid) sizes, positive associations for autozooid shape but no particular relationship between fecundity and brood chamber size. In addition, we demonstrate that long-term trait patterns are explained by palaeoclimate (as approximated by $\partial^{18}O$), and to a lesser extent by ecological interactions (i.e. overgrowth competition and substrate crowding). Our analyses show that macroevolutionary outcomes of trait evolution are not a simple scaling-up from the trait–fitness associations.

## 1. Background

Trait–fitness associations, especially when couched in terms of selection gradients [1], are traditionally studied on contemporary timescales [2,3]. Selection acts in concert with drift and constraints on changing ecological and environmental landscapes to produce phenotypes. There is robust documentation using wild, extant populations that directional phenotypic selection is strong in general [4] and stabilizing selection less common than expected [5]. These observations are puzzling [6] given the apparent long-term phenotypic stasis inferred from the fossil record [7]. This mismatch between our understanding based on microevolutionary studies of natural selection [4,5,8] and those attributed to macroevolutionary patterns of phenotypic change [7,9], must be bridged by not only further theory development but also data that allow us to peer directly into the deeper past. Such data can be provided by the fossil record [10].

Despite the potential of using the fossil record for studying phenotypes and fitness [11,12], few studies based on fossil data frame their questions around trait–fitness associations. Kurtén's [13] pioneering cross-sectional population study on Pleistocene cave bears compared age-at-mortality, a fitness component, with teeth morphology. Van Valen [14] followed in a conceptually similar study using Miocene horses, while others estimated relative fitness of morphotypes from one ontogenetic stage to the next, over millions of years of the evolutionary history of a group of crustaceans [15]. Using an exceptional study system, we assess, for the first time, the association between a fitness component (fecundity)

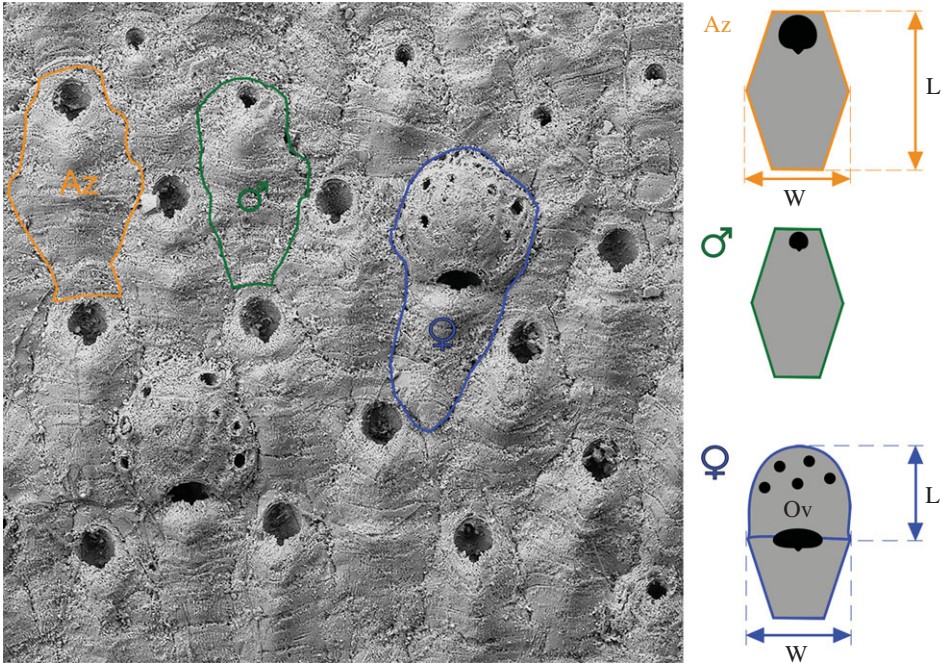

**Figure 1.** Scanning electron micrograph (SEM) of *Antarctothoa tongima* showing a typical section of a colony with examples of each polymorphic module outlined. Measurements we made are marked on the stylized figures of the same polymorphs (i.e. individuals in a colony with distinct morphologies and functions). This unregistered specimen from Nukumaru Limestone Formation is housed in the bryozoan palaeontological collection at the Natural History Museum London (NHMUK) (SEM micrograph number pdt18631). Az, autozooid (i.e. feeding polymorph, outlined in orange); Ov, ovicell (i.e. globular structure which serves as a brood chamber for a single larva). Female polymorphs including ovicells are outlined in blue, male polymorphs in green. L, length; W, width. (Online version in colour.)

and phenotypic traits for fossil populations of a single species in the deep past and ask if such associations are consistent with macroevolutionary patterns of the same phenotypes.

Here, we capitalize on the polymorphic, colonial nature of a metazoan group, namely cheilostome bryozoans, where we use the average density of female polymorphs within a genetic individual to estimate fecundity, which we use to approximate a component of fitness [16]. Our study system is the New Zealand cheilostome bryozoan *Antarctothoa tongima* (Ryland & Gordon, 1977). *Antartothoa tongima* is the ideal candidate for studying evolution on timescales of $10^4$–$10^6$ Myr because (i) key phenotypic traits are preserved and quantifiable from fossil colonies (figure 1; electronic supplementary material, figure S1); (ii) it has a rich fossil record spanning more than 2 Myr of turbulent climate history [17–19]; (iii) it is closely related to *Celleporella hyalina*, a species relatively well-studied in laboratory settings [20–22] on which we can root our assumptions.

Specific predictions for trait–fitness associations for three phenotypic traits, namely feeding polymorph (autozooid) size, brood chamber (ovicell) size and autozooid shape are as follows. Specifically, larger autozooid size has been found, across disparate cheilostome species, to be a key factor in increasing the probability of winning interspecific overgrowth competitions in spatial combats with other cheilostomes [18,23] (see electronic supplementary material, figure S2). Overgrowth has the effect of covering exit points (orifices) of feeding lophophores and hence cause partial to complete death of the overgrown colony. Whereas there is no large-scale trend in increasing autozooid size over the evolutionary history of cheilostome bryozoans despite apparent advantages, there is a tendency for ancestral species to give rise to descendent species that have larger autozooid sizes in the same lineage [24]. Hence, our naive prediction for *A. tongima* is that there should be a positive trait–fitness association for autozooid size. We also investigate the trait–fitness relationships of ovicell

size and the shape of autozooids (figure 1). Ovicell size is a strong predictor of larval and ancestrula (the parent module in a colony) sizes and hence of the survival chances of newly recruited colonies [25]. We hypothesize a trade-off between fecundity and ovicell size, while we did not, *a priori*, expect a relationship between fecundity and autozooid shape. We also expected all three traits to be relatively constant through 2 Myr, given the commonly observed phenotypic stasis in the fossil record [7].

In addition to studying trait–fitness associations, we also asked if each of the three traits and fecundity are detectably influenced by observed overgrowth competition and substrate crowding in our focal species, as such ecological interactions are known to negatively impact cheilostome bryozoans [26,27], and/or if palaeoclimatic conditions explain variation in these trait data.

## 2. Methods

### (a) Study system

*Antarctothoa tongima* of the family Hippothoidae, is a common encruster of hard substrates [28], living today off the coast of northern New Zealand. A pioneer species like others in the same family [29], *A. tongima* has a known depth range of *ca* 0–200 m [30]. It is commonly preserved in Pleistocene shellbeds cropping out near Wanganui and Waipukurau in the North Island of New Zealand [30]. In common with other hippothoids, *A. tongima* shows a degree of sexual dimorphism unusual among cheilostomes: within a colony, there is a complete separation between polymorphs having feeding and reproductive functions (figure 1; electronic supplementary material, figure S1). Female and male polymorphs have vestigial or rudimentary polypides lacking a gut, and consequently do not contribute to feeding [20,31].

We studied 414 fossil colonies of *A. tongima* collected in January 2014 and March 2017 from six Pleistocene formations of the Wanganui Basin, corresponding to six time intervals dated

from 2.29 to 0.30 Ma [32], with temporal durations ranging from 0.11 to 0.02 Myr (electronic supplementary material, table S1). The Wanganui Basin is a proto back-arc basin filled by a cyclic depositional sequence 2 km thick, spanning the last *ca* 2.5 Myr with a well-established, high-resolution chronostratigraphy [19,33,34]. To minimize environmental differences due to depositional factors, we targeted only shellbeds from transgressive system tracts [17,35], i.e. deposits that accumulated during coastal transgression. Note that the material we collected was not targeted at *A. tongima*. All the substrates we studied were bivalve shells.

## (b) Measurement of average fecundity and phenotypic traits

*Antarctothoa tongima* colonies are fragile and easily flake off the substrate, hence entire colonies are seldom preserved. We sampled groups of well-preserved zooids by taking light photographs using a ZEISS Stemi 508 stereoscope equipped with an Axiocam 105 colour camera. Each of these photographs is termed a 'spot' and is a spatially random sample of a single colony (electronic supplementary material, figure S1 and tables S1, S2). For each spot (*ca* 5 mm$^2$), we measured the area occupied by the colony and tabulated the number of autozooids (feeding polymorphs), males and females. Average fecundity is the density of females bearing ovicells per polymorph. Note that maximum area to which this species grows is about 300 mm$^2$ (DP Gordon 2020, personal communication) such that the minimum area we have sampled is 1.6% (if we only sampled one spot for a given colony).

The same images were used to measure the maximum length and width of autozooids and ovicells, a proxy for offspring (larval) size, using ImageJ2 [36]. Autozooids and ovicells were selected for measurements only if they were undamaged, clearly free from distortion due to the substrate or teratology, had well-defined boundaries and were astogenetically mature. We were able to make measurements for 311 distinct colonies (electronic supplementary material, table S2), recognized by the geometry and direction of growth. The median number of zooids measured per distinct colony is 13, but we note that within-colony morphometric variation can be captured by measurement of as few as three zooids [24]. Figure 1 shows how these traits were measured. We assured that the areas from which zooids and ovicells were measured were not tilted with respect to the frontal plane. Autozooid and ovicell sizes (i.e. areas) were estimated by multiplying maximum length by maximum width. For ovicell size, area estimated using the multiplication of length and width is highly correlated with that of a circle estimated using the average of length and width as radius ($R^2 = 0.988$; electronic supplementary material, figure S3), hence we only present results using the latter. For a subset of colonies, autozooids and ovicells were measured three times by the same person on different occasions to test the repeatability of our measurements. These measures are highly accurate between repeats ($R^2 > 0.97$; electronic supplementary material, figure S4). Both autozooid length and width predict autozooid area relatively well ($R^2 = 0.587$ and 0.719, respectively), but length and width predict each other poorly ($R^2 = 0.102$; electronic supplementary material, figure S5). Autozooid shape was obtained as maximum length divided by maximum width.

## (c) Estimating trait–fitness associations

We estimate a component of fitness as average fecundity of colonies (which are equivalent to genetic individuals in solitary organisms) in each of the six time intervals using counts of gravid females polymorphs (i.e. zooid with brood chambers/ ovicells that hold embryos) per polymorph unless otherwise stated. Note that the few female polymorphs not bearing ovicells (mean = 5.9%, median = 0% of all measured colonies with females

observed) were not included in the analyses, as they have not contributed to offspring by the time of death. Relative fitness is used as the response variable, and the (predictor) traits we investigate (i.e. autozooid area and shape and ovicell area) are natural logged. In other words, we have structured our analyses as one would for estimating multivariate selection gradients *sensu* Lande & Arnold [1], although we refrain from directly interpreting the coefficients as selection gradients as our data are time-averaged [37].

We analyse colony means of the three traits (i) in multivariate analyses for the entire 2 Myr as a whole (*N* = 169, where no trait data are missing), and (ii) in univariate analyses in each time interval where each trait is represented by more data (*N* = 230, 244, 230 for autozooid area, ovicell area and autozooid shape, respectively). We use binomial glms to model fecundity (number of ovicells given the total number of polymorphs, presented in figures 3 to 5). Alternatively, we also used a Poisson glm to model the number of ovicells, but using the area on which these are observed as a covariate (see electronic supplementary material, tables S6, S7 and figures S12–S15). We also present linear models where we use the ratio of ovicells to total number of polymorphs as the response variable (see electronic supplementary material, tables S8, S9 and figures S16–S19) because selection gradients were formally cast as such [1]. Model diagnostics are presented in the R Markdown file.

We systematically explored the relationship between trait means (time interval averages) and the strength of the association in all the univariate trait–fitness associations we performed (see above) because when plotting non-standardized trait values we observed that some trait–fitness associations are weaker when traits are smaller (e.g. figure 4).

## (d) Characterizing trait variance and the effect of time-averaging

To gain understanding of the trait variability within colonies (i.e. variation in trait given the same underlying genotype = 'plasticity'), across colonies (variation due to genetic differences), and across time intervals (variation due to longer term evolution), we plotted the variances of the three traits for each colony, as well as for colonies grouped by the time interval (electronic supplementary material, figures S6–S9). As one might suspect that lumping samples from a longer time interval might inflate trait variance, we inspected the relationship between trait variance and the duration of the given time interval (electronic supplementary material, figure S10). In addition, we compare the variances of larger, but temporally constrained samples within a time interval, to smaller but temporally more dispersed samples, from the same time interval (electronic supplementary material, table S3).

## (e) Response of phenotypic traits and fecundity to overgrowth competition, crowding and palaeoenvironment

We use simple linear models to study the relationship between each of our three (normally distributed) logged traits with (i) the number of observed inter-/intraspecific overgrowth competition (electronic supplementary material, figure S2) already documented in previous publications [18,23,35] or collected for the purpose of the current study, (ii) the total number of bryozoan colonies observed on the same substrate (used as an estimate for crowding as a stress factor that may affect fecundity and phenotypic traits), and (iii) the mean and standard deviation of the $\partial^{18}O$ values in each time interval (data from [38]), which mostly reflects seawater temperature, commonly used as a proxy for environmental change and overall climate state (e.g. [7]).

We used a binomial glm to study the relationship between fecundity (as a response variable), the three traits, overgrowth, crowding and palaeoclimate (as predictor variables). In each

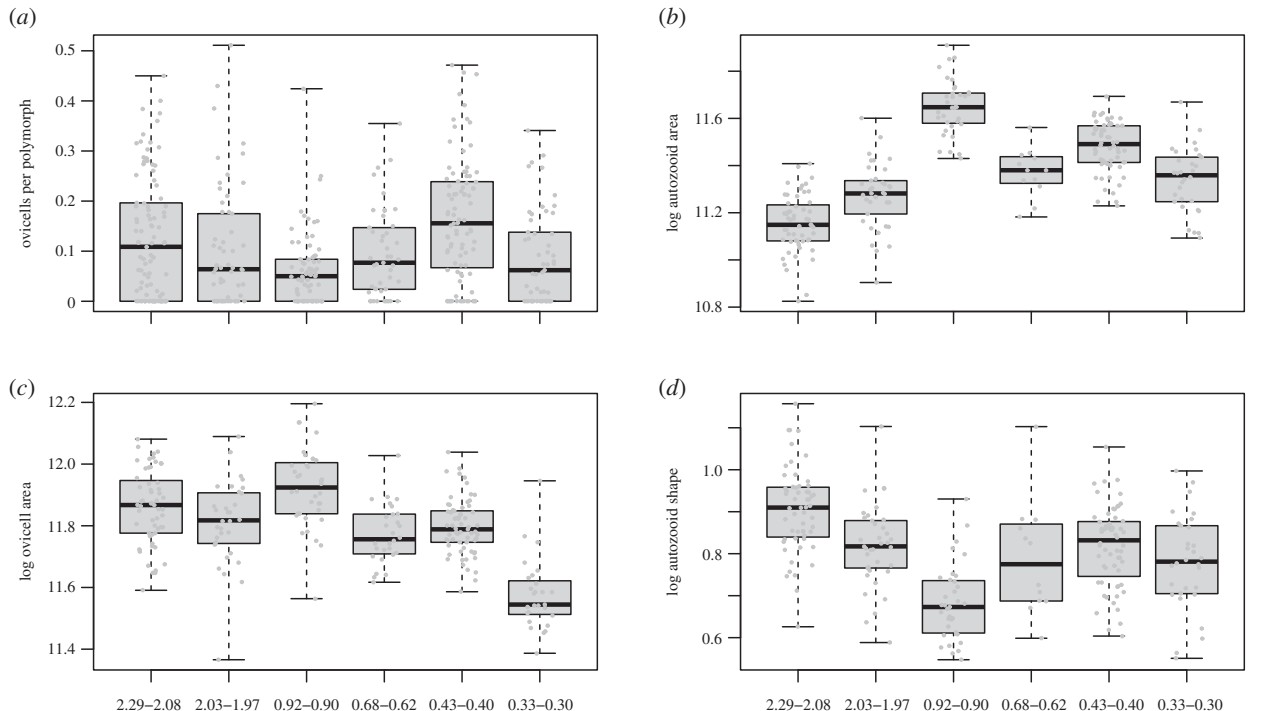

**Figure 2.** Temporal changes for *A. tongima* colonies over 2 Myr. The six time intervals are plotted with medians, the boxed interquartile ranges and the span of the data. (*a*) Plots the number of gravid females (ovicells) per polymorph, (*b*) autozooid and (*c*) ovicell area, which are both in natural log μm²; and (*d*) autozooid shape which is dimensionless, but also natural logged. The data (colony averages or values) are grey points on the boxplots, with scatter for visibility. Numbers below the second row show the age ranges in million years of Nukumaru Limestone, Nukumaru Brown Sand, Lower Kai-iwi Shellbed, Upper Kai-iwi Shellbed, Shakespeare Cliff Basal Sand Shellbed and Landguard Formation, respectively.

case, models are compared using AIC. All analyses were run in R. v. 3.6.1 [39].

## 3. Results

### (a) Phenotypic traits and fecundity show substantial variation within and across time intervals

The three focal traits, autozooid size (area approximated by length times width), ovicell size (area approximated by $\pi r^2$ where r is the average of ovicell length and width), and autozooid shape (length divided by width, figure 1) for *A. tongima* colonies vary within time intervals and through time, as does fecundity, estimated by the number of gravid females (i.e. females bearing ovicells) per polymorph (figure 2).

Within-colony trait variance is smaller than trait variance at population level (i.e. within each formation) for all three traits (electronic supplementary material, figure S6). Likewise, trait variance within each formation is smaller than that for all formations combined (electronic supplementary material, figures S7–S9). Although there is time-averaging within each formation, we show that the variance of samples (very short time windows in a formation) represented by multiple colonies are not distinguishable from the variance of time-averaged colonies (the time window of the whole formation, see electronic supplementary material, table S3).

### (b) Trait–fitness associations are relatively stable through time

Using a data subset of *N* = 169 colonies, with no missing data for any of the three traits, the best AIC-ranked multivariate binomial generalized linear model (glm) of colony-level relative fitness

(electronic supplementary material, table S4), estimated as fecundity, suggests a negative trait–fitness relationship for log autozooid area, no relationship for log ovicell area and a positive trait–fitness relationship for skinnier (i.e. relatively longer) autozooids (figure 3; electronic supplementary material, table S5). By estimating these trait–fitness associations for the six time intervals using a univariate binomial glm for each trait separately, we found that the first relationship is negative in five of six time intervals (*N* = 230; figure 4), the second has no detectable relationship in five of six time intervals (*N* = 244; electronic supplementary material, figure S11), and the third has a positive relationship in four of six time intervals (*N* = 230; figure 5). These results are robust to alternative approximations of fecundity and models with different assumptions of response variable (relative fitness) error distributions (see electronic supplementary material, tables S6–S9 and figures S12–S19 for linear and Poisson glm results).

We observed that the negative trait–fitness association for autozooid area is absent when *A. tongima* is at its smallest in our dataset (Nukumaru Limestone Formation 2.29–2.08 Ma; figure 4). This association tends to be stronger when autozooid areas are large, although there is no statistical correlation between the strength of this association and average autozooid areas (electronic supplementary material, figure S20). For completeness, we also explored the correlation between the strength of the trait–fitness relationship and the mean of the traits involved for ovicell area and autozooid shape, but found no correlation (electronic supplementary material, figures S21–S28).

### (c) $\partial^{18}$O, but not overgrowth competition and crowding, effects phenotypic traits and fecundity

The best (linear) model for autozooid area (electronic supplementary material, table S10), while including overgrowth

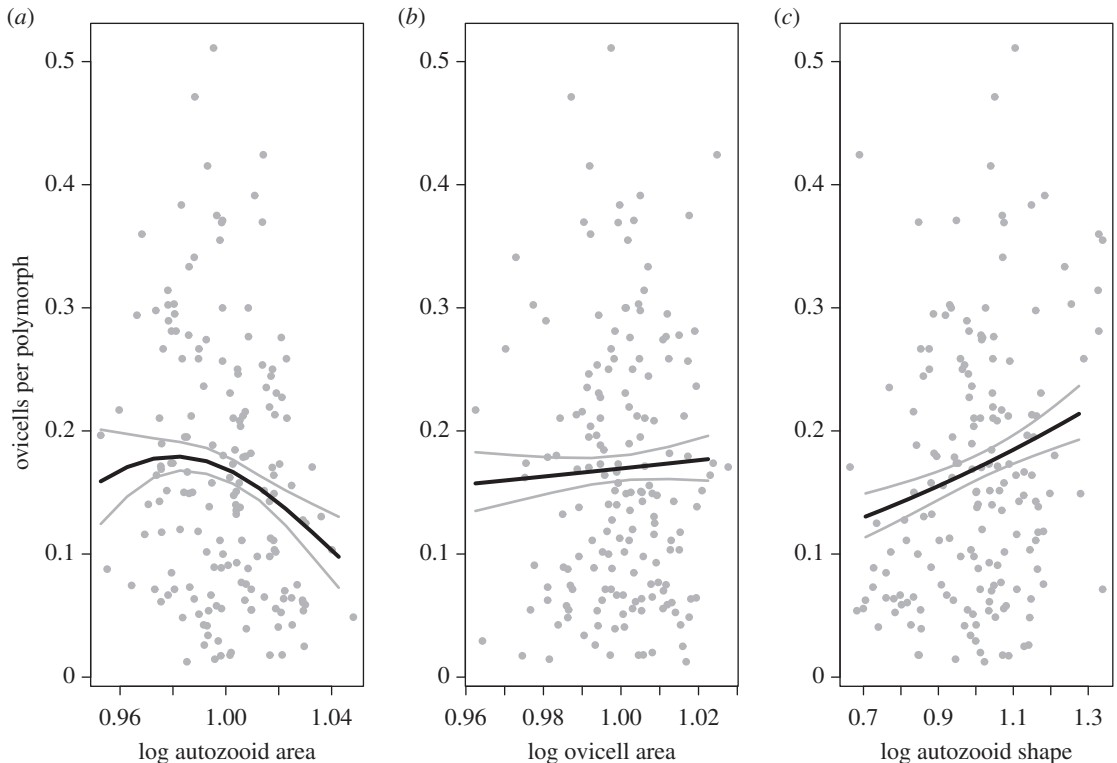

**Figure 3.** The panels show fitted binomial (glm) multivariate trait–fitness relationships from the best AIC-ranked model (electronic supplementary material, table S1), combining data from all time intervals ($N = 169$). Relative fitness (fecundity) is approximated with the number of gravid females (i.e. ovicells) per polymorph, and the values on the x axes are natural logged and mean standardized. (a) Plots autozooid area (originally measured in $\mu m^2$); (b) ovicell areas (originally measured in $\mu m^2$) and (c) autozooid shape, which is dimensionless. Grey dots are colony-level averages, black lines are predicted values and grey lines the 95% CI. Parameter estimates from this model are given in electronic supplementary material, table S2.

interactions (electronic supplementary material, figures S2, S29), shows only a weak effect of crowding (i.e. total number of bryozoan colonies observed on the same substrate), but a clear indication that $\partial^{18}O$, a proxy for palaeoclimate, has a positive relationship with autozooid size where one unit change in $\partial^{18}O$ is predicted to result in $0.213 \pm 0.049$ log units of change in size (figure 6; electronic supplementary material, table S11). The best linear model for ovicell area (electronic supplementary material, table S12) indicates that variability in $\partial^{18}O$ has a negative effect and intraspecific overgrowth interactions have a positive effect (electronic supplementary material, table S13). The best linear model for autozooid shape (electronic supplementary material, table S14) features overgrowth interactions and $\partial^{18}O$ both having estimated small or highly uncertain effects (electronic supplementary material, table S15). Fecundity is best explained using a binomial glm excluding overgrowth interactions (electronic supplementary material, table S16). However, this model demonstrates a strong effect of $\partial^{18}O$, where one unit change in $\partial^{18}O$ increases the odds of ovicells by $0.397 \pm 0.119$ units (electronic supplementary material, table S17).

## 4. Discussion

Fitness is a slippery yet crucial concept in evolutionary biology [40], and one that undoubtedly, but rarely, benefits from insights from the fossil record. The strength of this work lies in the unique possibility of estimating fecundity, a fitness component, in fossil colonies of *A. tongima*. Fecundity was used to capture the dynamics of trait–fitness associations for three traits in six time intervals spread over 2 Myr of

evolution of a colonial marine species, going beyond the temporal constraints of studies of contemporary populations spanning several decades [3,41]. In the following discussion, we use the term 'fit' when interpreting results based on our estimated fecundity proxy, but note that this interpretation should be made cautiously (see also caveats section).

We demonstrated that the trait–fitness associations can be relatively stable on geological timescales (figures 4 and 5), enlightening results from short-term contemporary studies [8,42]. Yet the resulting phenotypic change (figure 2b,d) sometimes appears to contradict predictions based strictly on trait–fitness associations. We have also shown substantial change of our studied traits over time, contradicting the apparent phenotypic stasis often observed in the fossil record.

Take the example of autozooid size, a heritable trait in cheilostome bryozoans [43]. Smaller autozooid size is associated with higher fecundity (figure 3) in four of six observable snapshots in the history of *A. tongima* (figure 4), contrary to our naive predictions. Why are colonies with smaller autozooids fitter, in the sense that they can have more offspring at any one time, judging by the density of ovicells? Smaller and consequently more densely packed autozooids increase the number of feeding organs (lophophores) to surface area [44]. More efficient feeding, in turn, speeds up the accumulation of resources to allow for reproduction [45]. However, the relationship between autozooid size and lophophore size/number/efficiency has not yet been quantified as far as we are aware. More importantly, smaller autozooid size enables faster physiological sharing of resources among modules within colonies [46,47]. However, despite higher fecundity for colonies with smaller autozooid size, these sizes did not systematically decline over evolutionary history (figure 2b).

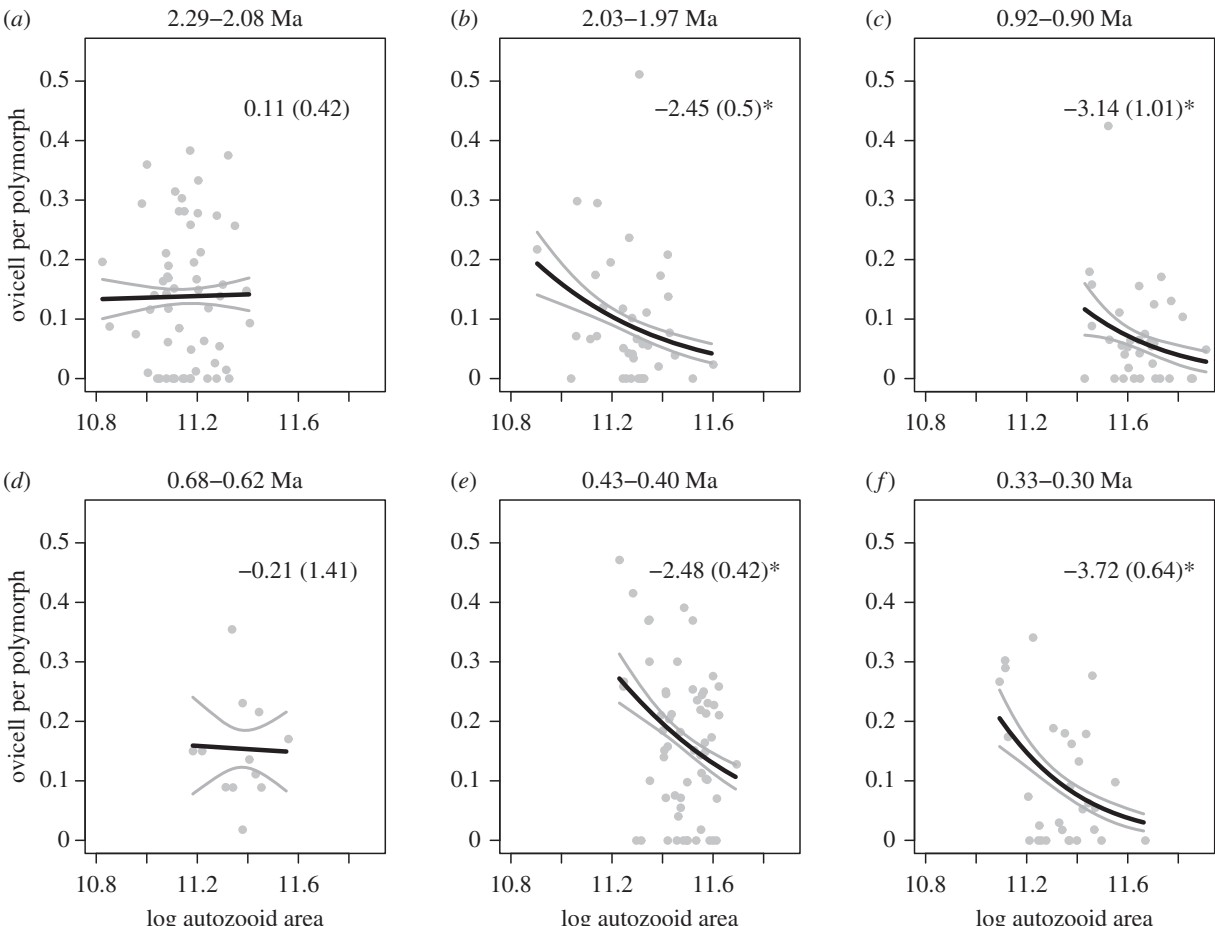

**Figure 4.** The panels show negative trait–fitness relationships for autozooid area. The average number of gravid females (ovicell) per polymorph are plotted against average log autozooid areas ($\mu m^2$) for each colony (grey dots, $N = 230$). Time intervals are indicated in Ma (=millions of years ago) on top of each panel. Solid black and grey lines show the prediction using a univariate binomial glm, and their 95% CI, respectively. Numbers within each panel are predictions with their standard errors in parentheses. Four estimates are significantly negative at a $p < 0.05$ level (indicated by *).

Instead, it is apparent that part of the temporal autozooid size variation can be explained by changing palaeoclimatic conditions (i.e. seawater temperature as approximated by $\partial^{18}O$; figure 6; electronic supplementary material, tables S10 and S11), reminiscent of the simulation results from [7]. In fact, a similar temporal pattern of autozooid size change is shown by other cheilostome species in the Wanganui Basin over the same time interval [18]. The multi-species, temporally concerted phenotypic pattern is likely in part driven by external temperatures at the time the zooids were built, where lower temperatures produce larger autozooids [48]. We hypothesized that colonies with larger autozooids may also have a fitness advantage as it has been observed that species with larger autozooids are more likely to win, and hence survive, overgrowth interactions [18]. However, no significant influence of overgrowth interactions on autozooid size was detected for *A. tongima* (electronic supplementary material, tables S11).

Colonies with skinnier autozooids also seem to be fitter in general, although there is a reversal of this trait–fitness association in one of the time intervals (figure 5; electronic supplementary material, figures S15 and S19). Skinnier autozooids may accommodate polypids with relatively longer tentacles, which may translate into more efficient feeding per autozooid [49], and hence more resources for reproduction. The shape trait is also positively influenced by the observed number of intraspecific overgrowth interactions (electronic

supplementary material, tables S14 and S15), a result we did not anticipate. Here, we might speculate that skinnier autozooids are a reflection of higher growth rates especially when meeting conspecifics to increase the chance of contact and fusion between colonies [50]. This fusion (homosyndrome) can increase survival and reproduction for the adjoined colonies and occurs more often in poor competitors [51], such as *A. tongima* [29].

Finally, the trade-off between ovicell (i.e. offspring) size and number, commonly observed among solitary organisms [52], was not detectable in our data for *A. tongima* (electronic supplementary material, figure S18). Intriguingly, intraspecific interactions are associated with larger ovicell size, suggesting that intraspecific interactions could contribute positively to reproduction (electronic supplementary material, table S13).

As in all studies, we have caveats to consider. We have assumed that samples from each formation are a fair representation of the populations that existed during those time intervals. The six formations, while deposited in similar environments [17,35], differ in temporal duration. However, within-sample trait variances (representing a relatively short amount of time) are not distinguishable from those resulting from multiple samples (representing a greater amount of time) within a formation (electronic supplementary material, table S3). This low inflation of variance due to time-averaging in the fossil record, also documented in other data [37,53], gives us some confidence in assuming that *A. tongima* samples

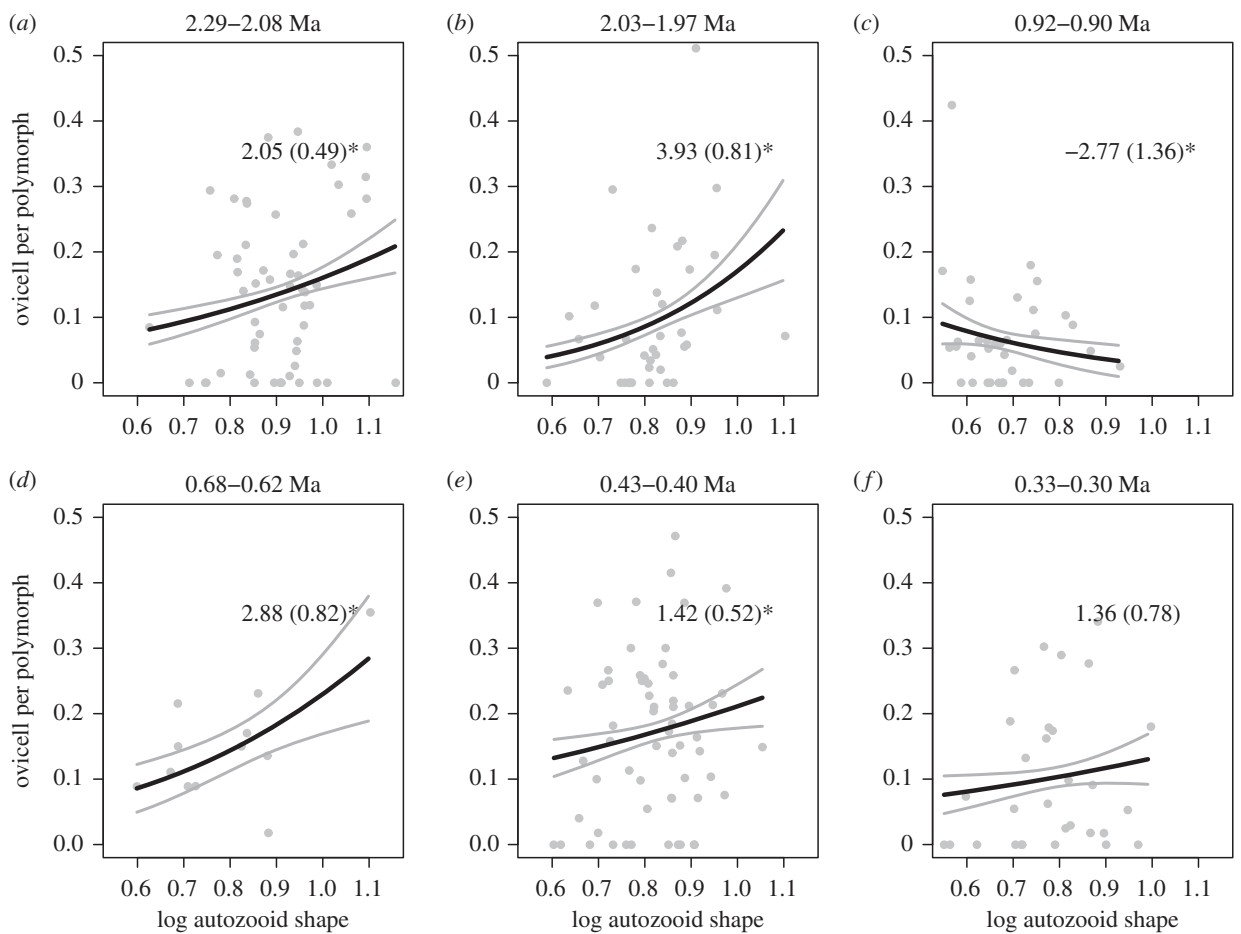

**Figure 5.** The panels show positive trait–fitness relationships for autozooid shape. The average number of gravid females (ovicells) per polymorph are plotted against average log autozooid shape (dimensionless) for each colony (grey dots, $N = 230$). Time intervals are indicated in Ma (=millions of years ago) on top of each panel. Solid black and grey lines show the prediction using a univariate binomial glm, and their 95% CI, respectively. Numbers within each panel are estimates with their standard errors in parentheses. Five estimates are significantly positive at a $p < 0.05$ level (indicated by a *).

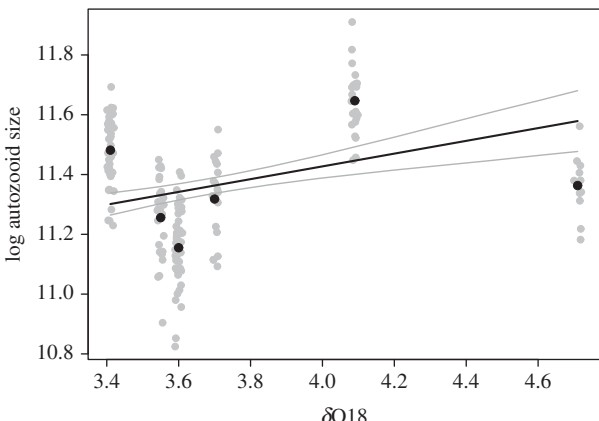

**Figure 6.** Relationship between $\partial^{18}O$ and log autozooid size. Grey dots are average log autozooid size for colonies given mean $\partial^{18}O$ value for each of the six formations, plotted with a jitter for visibility. Black dots are log autozooid means for each time interval. The black line is the prediction when other variables in the model are held constant and the grey its 95% CI (see electronic supplementary material, table S8 for details).

from different formations are comparable. We based the species identies of examined specimens only on their skeletal features. While constrained by our fossil material, we note that skeletal features, as experimentally shown in several distantly related cheilostome species, correspond well with genetically identified species [43]. We have also assumed that the distinct colonies studied are genetically independent.

Yet, if they are preserved on the same substrate or within the same sample, they may not be. However, it is known from microsatellite studies of the closely related *C. hyalina* that larvae settle randomly with respect to kin [54]. We used one to several small random 'spots' of preserved colonies to estimate morphological traits and fecundity (electronic supplementary material, tables S1 and S2), but note that much of the within-species variation is captured by only measuring few zooids within few colonies [24]. We also assumed (see electronic supplementary material, figures S7–S9) that we have captured much of the 30–50% of trait variation that is external to within-colony variation [55]. We used the density of females bearing ovicells as a fitness component as commonly done in cheilostome life-history studies [16]. However, assuming that ovicell density linearly translates to successful offspring may cause an overestimation of fecundity. On the other hand, this proxy may also be an underestimate as some cheilostome species can reuse their ovicells [56], although ovicell recycling has never been demonstrated in *A. tongima*. In some cheilostomes, females are spatially structured, such that random samples of colony sections may be biased for estimates of average fecundity. However, in *Antarctothoa*, the placement of female polymorphs are distributed haphazardly [57]. The number of overgrowth interactions we observed in our data must be an underestimate, as parts of colonies (and their competitors) can break-off their substrate both before and after fossilization. Despite this, observed overgrowth interactions should be random with respect to

our three traits and fecundity. We used $\partial^{18}O$ values averaged over the whole duration of each formation, while coarse, is the best estimate we have for the palaeoenvironment. Phenotypic plasticity [58] is present among the colonies but is smaller than the variation that is seen across the samples (electronic supplementary material, figures S6–S9). In addition, we acknowledge that we have only estimated one component of fitness as done in most studies based on natural populations (see examples in [8]), namely fecundity, while lifetime fitness is also dependent on survival and growth. Finally, any other variables not explicitly considered because they cannot be controlled for (e.g. colony growth rate, rate of fertilization) are assumed not to be systematically varying across our temporal comparisons and hence do not bias inferences we make on the phenotypic traits investigated, a common practice in selection studies in the wild [8].

## 5. Conclusion

We captured nuances in the dynamics of trait evolution given changing environmental and ecological conditions over geological timescales and corroborated the lack of phenotypic stasis in the fossil record found in recent work [59]. While previous studies investigate life-history traits [60] and pioneered quantitative genetics using fossil organisms [61,62], this is the first attempt at quantifying trait–fitness associations and understanding how they align with long-term phenotypic evolution, given the backdrop of climate change and ecological interactions. Although 'small is fit' within bouts of time presented,

'smaller and smaller' was not the resulting phenotypic pattern. Rather, macroevolutionary outcomes are a complex interplay of the constraints of physiological functions and the changing biotic and abiotic environments. These results will open doors to integrating insights from the fossil record with microevolutionary theory to further bridge empirical and conceptual gaps between micro- and macroevolution.

Data accessibility. The code and data necessary to reproduce our analyses, including model diagnostics, are supplied as a word file (Antarctothoa.Rmd.3.11.2020.docx) and an excel file (Antarctothoa_dataset_07.08.2019.xlsx), while $\partial^{18}O$ data from Lisieki & Raymo (2005) are supplied as a csv file ($\partial^{18}O$.csv). Data and code are available from the Dryad Digital Repository: https://doi.org/10.5061/dryad.dz08kprw9 [63].

Authors' contributions. E.D.M. and L.H.L. came up with the study; E.D.M. collected the bulk of the data and made figure 1, electronic supplementary material, figures S1 and S2; L.H.L. did the analyses, L.H.L. wrote the first draft of the manuscript that E.D.M. and L.H.L. revised together.

Competing interests. We declare we have no competing interests.

Funding. This work was supported by the European Research Council (ERC) under the European Union's Horizon 2020 research and innovation programme (grant agreement no. 724324 to L.H.L.).

Acknowledgements. We are grateful to two anonymous reviewers whose comments helped improving our writing. We thank Alan Beu, Jeoren Boeve, Emily Enevoldsen, Dennis P. Gordon, Seabourne Rust, Carolann Schack, Paul D. Taylor and Kjetil L. Voje for their assistance in the field and GNS for collecting and export permits. We also thank Mali H. Ramsfjell for assistance in the laboratory, Paul D. Taylor for making his SEMs available. We thank Mark Grabowski, David Houle, Michael Morrissey, Arthur Porto, Trond Reitan, Paul. D. Taylor, Yngvild Vildenes and Kjetil L. Voje for discussions at various stages of this study.

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
