## [Reviewer comments · Proceedings of the Royal Society B: Biological Sciences]

Review History

RSPB-2020-2047.R0 (Original submission)

Review form: Reviewer 1

Recommendation

Major revision is needed (please make suggestions in comments)

Scientific importance: Is the manuscript an original and important contribution to its field?

Good

General interest: Is the paper of sufficient general interest?

Excellent

Quality of the paper: Is the overall quality of the paper suitable?

Good

Is the length of the paper justified?

No

Should the paper be seen by a specialist statistical reviewer?

Yes

Do you have any concerns about statistical analyses in this paper? If so, please specify them explicitly in your report.

Yes

It is a condition of publication that authors make their supporting data, code and materials available - either as supplementary material or hosted in an external repository. Please rate, if applicable, the supporting data on the following criteria.

Is it accessible?

Yes

Is it clear?

Yes

Is it adequate?

No

Do you have any ethical concerns with this paper?

No

Comments to the Author

This is a very impressive study. The research questions and context are important and the data and methods have the potential to address them. I believe that it could be an even stronger paper if a couple of key points were addressed, as well as some minor ones for clarity.

Is colony wide variation adequately accounted for by measurements from a portion of the colony? In multiple separate studies, Cheetham, Hughes, Hageman and coauthors for each, have regularly documented that about 40-60% of variation for a morphometric character within a bryozoan species (or at least population/assemblage) accounted for by within colony (non-genetic) variation. These, combined and independent studies of O'Dea and Okamura and their coauthors, also document positional effects within colonies, both at the level of patches/spots within colonies and nearest neighbor effects of adjacent/nearby zooids.

The authors of the present study need to comment on how emphasis of measurements concentrated at the spot level within a colony (~76% of colonies represented by measurement from a single ~5mm² "spot"), which represents a biased subset of the total within colony variation. Is one spot the size of a typical who colony, or what proportion of a mature colony would a spot/fragment represent. These factors may not be relevant to conclusions of the paper, but the rationale for disregarding them, should be addressed logically and stated clearly. Figures S6-9 attempt to constrain this, but the conclusions, implications taken from these figures needs to be better explained in the text. That is, do the authors believe that these results alleviate the concerns about measurements from a limited number of spots per colony adequately representing variation within colonies?

By the end the paper, "fitness" and "fit" are treated as literal synonyms for ovicell size and number. I expect there is some truth in that conclusion, but the paper does not get far enough beyond the tautology to prove this. The logic that the number and or size of ovicells is a proxy for fecundity and that fecundity is related to (measure of?) fitness is elegant and simple. The literature on fitness both theoretic and measured is rich, but I adhere to the idea that the definition of fitness is an ever-moving evolutionary target. If larval recruitment between two taxa in a given setting, an ovicelled and non-ovicelled taxon, then in that circumstance, ovicell number is not a meaningful metric of relative fitness among species, so how does one demonstrate that it is within species? Overgrowth "strategies" and fitness are more directly observed and evaluated, but even there the end game of "lesser fitness" has not resulted in the extinction of the cyclostomes. At the very least one could argue that some non-ovicellate taxa thrive in certain settings with alternative definitions of "fitness".

The authors do not need to more fully demonstrate the biological fitness of a character trait, but I encourage them to be mindful that they are still working in a complex system and have not proven the overall fitness of an organism based on a single proxy. I am sure the author's would agree, but again, I encourage them to avoid the tone that they have definitively defined fitness in this clade.

Other questions for clarification.

Line 105, Table S1- Clarify the methods used to calculate the temporal resolution. Probability statistic or just serial ranges?

Line 117 - Approximately how much area of a mature colony does a single "spot" represent?

Line 125 - Clarify that two spots off the same shell definitely represent two colonies (genotypes), i.e., these 311 groups are separate colonies (genotypes), not spots of which could be parts of a larger colony (shared genotype).

Line 125 - Clarify that two spots off the same shell definitely represent two colonies (genotypes), i.e., these 311 groups are separate colonies (genotypes), not spots of which could be parts of a larger colony (shared genotype).

Line 268 - It would be useful to see a calculation of water volume clearance for small vs. large lophophores, both at closest packing. Water volume clearance per time is the key, not just area.

Line 268 - reword, "why do colonies with more & larger ovicells have smaller autozooids?"

Line 295 - is this just canalization from packing constraints, or some other controlling factor?

Line 275 - are the authors calling on ambient temperature as the control (environmental) of the autozoid size phenotype? or calling on evolutionary changes to the genotype as a fitness response?

Line 276 - Do not use the implied content of citation "[7]" as the subject in the sentence.

Line 128 - What is the "downstream" analysis?

Line 134 - Please clarify, is "highly accurate" measurement replication a function of single researcher's replicates or replication among multiple workers measuring the same material? (typically a big difference).

Review form: Reviewer 2

Recommendation

Accept with minor revision (please list in comments)

Scientific importance: Is the manuscript an original and important contribution to its field?

Excellent

General interest: Is the paper of sufficient general interest?

Excellent

Quality of the paper: Is the overall quality of the paper suitable?

Excellent

Is the length of the paper justified?

Yes

Should the paper be seen by a specialist statistical reviewer?

No

Do you have any concerns about statistical analyses in this paper? If so, please specify them explicitly in your report.

Yes

It is a condition of publication that authors make their supporting data, code and materials available - either as supplementary material or hosted in an external repository. Please rate, if applicable, the supporting data on the following criteria.

Is it accessible?

Yes

Is it clear?

Yes

Is it adequate?

Yes

Do you have any ethical concerns with this paper?

No

Comments to the Author

The present work analyses the correlation between fitness (measured as fecundity) and other traits in the deep time. The study system is excellent since few taxa have good enough fossil material for the proposed aim, added to work on living relatives that support the hyp/assumptions presented. The study is of interest to the broad readership of Proceedings B.

I recommend publication and have only minor comments/suggestions before the paper can be published.

Abstract

Line 24 - paleoclimate is too general to be informative in this context, I think. Maybe clarify which proxy used to reconstruct past climate variation. Ecological interaction is a broad term, interspecific competition or overgrowth seems clearer.

Background

Line 51 - component of fitness measured as average female fertility should be clearly stated right away.

Moreover, terms fitness/fertility appear sometimes alone, sometimes together (i.e., fitness (fecundity)) in the next sections and somehow that felt more confusing than helpful. Consistency may help.

Line 57 - following the previous suggestion 'a fitness component' can be removed.

Line 64 - would be great to see the hyp/predictions listed clearly (instead of point iv). The next paragraph would explain the rationale behind each hyp/prediction. This would help the reader follow the different tests/hyp later on.

Line 68 - this sentence repeats the idea in Line 51-53 correct?

Line 72 - S is missing in figure S2

Methods

Line 116 - how can you tell it is one colony per spot?

Line 139 to 144 - Maybe this paragraph should start subsection (e).

Plus overgrowth of colonies = interspecific competition? Again, for someone not very familiar with the system it is unclear what you mean right away.

Line 150 to 151 - odd sentence. Maybe change into something as 'The few female polymorphisms identified without ovicells were not accounted in the analyses (mean = (...))'.

Line 165, 166 - it reads (see electronic supplementary material) but no Suppl table/fig is indicated.

General comment Methods

From figure 2 legend and table S1 each time interval corresponds to a formation. Yet this is not clear in the text.

The time intervals (formations) have distinct spans. Also, the number of spots/samples is uneven. Overall, the authors do a good job of covering issues related to this. Same with dealing with environmental differences resulting from sample origin as well as preservation.

Results

Line 201 to 203 - how you obtain size or shape for the three traits is much clearer here than in the Methods section.

Moreover, size is used in some sections and area in the others. E.g., line 128 - 'Autozooid area we use for downstream analysis (...)'; line 201 - 'autozooid size (area approximated by length times width)'. Maybe use area (shape) across and when defining how you estimate this use a system closer to the Results format? Seems straight forward but on a first read, consistency helps...

Line 232 - crowding discussed but not in the title.

Discussion

Line 271 - where it reads 'more rapid' change to 'faster'.

Line 276 - missing punctuation after [7].

Line 294 - do you mean when fusion occurs new autozooids are narrower?

Line 308 to 310 - confusing sentence. Do you mean within-sample variance (colonies from substracts collected in the same location) is similar to variance resulting from multiple samples within a formation?

Line 313 to 319 - may be more appropriate in the Methods section. Plus cuts the Discussion flow a bit, which is a pity.

Figures

Fig 5 - legend indicates 4 estimates (word misspelt in legend) are significantly positive but 5 * are shown.

FigS10 - highlight Nukumaru Limestone formation so the note comment can be seen.

Decision letter (RSPB-2020-2047.R0)

26-Oct-2020

Dear Dr Di Martino:

Your manuscript has now been peer reviewed and the reviews have been assessed by an Associate Editor. The reviewers' comments (not including confidential comments to the Editor) and the comments from the Associate Editor are included at the end of this email for your

reference. As you will see, the reviewers and the Editors have raised some concerns with your manuscript and we would like to invite you to revise your manuscript to address them.

Research ethics:

Use of animals and field studies:

It is a condition of publication that you make available the data and research materials supporting the results in the article. Please see our Data Sharing Policies (<https://royalsociety.org/journals/authors/author-guidelines/#data>). Datasets should be deposited in an appropriate publicly available repository and details of the associated accession number, link or DOI to the datasets must be included in the Data Accessibility section of the article (<https://royalsociety.org/journals/ethics-policies/data-sharing-mining/>). Reference(s) to datasets should also be included in the reference list of the article with DOIs (where available).

Please submit a copy of your revised paper within three weeks. If we do not hear from you within this time your manuscript will be rejected. If you are unable to meet this deadline please let us know as soon as possible, as we may be able to grant a short extension.

Best wishes,
Dr John Hutchinson, Editor
mailto:proceedingsb@royalsociety.org

Associate Editor
Comments to Author:
Dear Authors

You will see that two reviewers have found the study of great interest and recommend publication. However, a number of major and minor comments are raised that could improve the study. In particular reviewer 1 raises some comments about intra-colony variability and the sampling of a colony, which seems necessary to explicitly clarify the nature of.

Reviewer(s)' Comments to Author:

Referee: 1

Comments to the Author(s)

This is a very impressive study. The research questions and context are important and the data and methods have the potential to address them. I believe that it could be an even stronger paper if a couple of key points were addressed, as well as some minor ones for clarity.

Is colony wide variation adequately accounted for by measurements from a portion of the colony? In multiple separate studies, Cheetham, Hughes, Hageman and coauthors for each, have regularly documented that about 40-60% of variation for a morphometric character within a bryozoan species (or at least population/assemblage) accounted for by within colony (non-genetic) variation. These, combined and independent studies of O'Dea and Okamura and their coauthors, also document positional effects within colonies, both at the level of patches/spots within colonies and nearest neighbor effects of adjacent/nearby zooids.

The authors of the present study need to comment on how emphasis of measurements concentrated at the spot level within a colony (~76% of colonies represented by measurement

from a single ~5mm² “spot”), which represents a biased subset of the total within colony variation. Is one spot the size of a typical whole colony, or what proportion of a mature colony would a spot/fragment represent. These factors may not be relevant to conclusions of the paper, but the rationale for disregarding them, should be addressed logically and stated clearly. Figures S6-9 attempt to constrain this, but the conclusions, implications taken from these figures needs to be better explained in the text. That is, do the authors believe that these results alleviate the concerns about measurements from a limited number of spots per colony adequately representing variation within colonies?

By the end the paper, “fitness” and “fit” are treated as literal synonyms for ovicell size and number. I expect there is some truth in that conclusion, but the paper does not get far enough beyond the tautology to prove this. The logic that the number and or size of ovicells is a proxy for fecundity and that fecundity is related to (measure of?) fitness is elegant and simple. The literature on fitness both theoretic and measured is rich, but I adhere to the idea that the definition of fitness is an ever-moving evolutionary target. If larval recruitment between two taxa in a given setting, an ovicelled and non-ovicelled taxon, then in that circumstance, ovicell number is not a meaningful metric of relative fitness among species, so how does one demonstrate that it is within species? Overgrowth “strategies” and fitness are more directly observed and evaluated, but even there the end game of “lesser fitness” has not resulted in the extinction of the cyclostomes. At the very least one could argue that some non-ovicellate taxa thrive in certain settings with alternative definitions of “fitness”.

The authors do not need to more fully demonstrate the biological fitness of a character trait, but I encourage them to be mindful that they are still working in a complex system and have not proven the overall fitness of an organism based on a single proxy. I am sure the author’s would agree, but again, I encourage them to avoid the tone that they have definitively defined fitness in this clade.

Other questions for clarification.

Line 105, Table S1- Clarify the methods used to calculate the temporal resolution. Probability statistic or just serial ranges?

Line 117 - Approximately how much area of a mature colony does a single “spot” represent?

Line 125 - Clarify that two spots off the same shell definitely represent two colonies (genotypes), i.e., these 311 groups are separate colonies (genotypes), not spots of which could be parts of a larger colony (shared genotype).

Line 125 - Clarify that two spots off the same shell definitely represent two colonies (genotypes), i.e., these 311 groups are separate colonies (genotypes), not spots of which could be parts of a larger colony (shared genotype).

Line 268 - It would be useful to see a calculation of water volume clearance for small vs. large lophophores, both at closest packing. Water volume clearance per time is the key, not just area.

Line 268 - reword, “why do colonies with more & larger ovicells have smaller autozooids?”

Line 295 - is this just canalization from packing constraints, or some other controlling factor?

Line 275 - are the authors calling on ambient temperature as the control (environmental) of the autozoid size phenotype? or calling on evolutionary changes to the genotype as a fitness response?

Line 276 - Do not use the implied content of citation “[7]” as the subject in the sentence.

Line 128 - What is the “downstream” analysis?

Line 134 - Please clarify, is “highly accurate” measurement replication a function of single researcher’s replicates or replication among multiple workers measuring the same material? (typically a big difference).

Referee: 2

Comments to the Author(s)

The present work analyses the correlation between fitness (measured as fecundity) and other traits in the deep time. The study system is excellent since few taxa have good enough fossil material for the proposed aim, added to work on living relatives that support the hyp/assumptions presented. The study is of interest to the broad readership of Proceedings B.

I recommend publication and have only minor comments/suggestions before the paper can be published.

Abstract

Line 24 - paleoclimate is too general to be informative in this context, I think. Maybe clarify which proxy used to reconstruct past climate variation. Ecological interaction is a broad term, interspecific competition or overgrowth seems clearer.

Background

Line 51 - component of fitness measured as average female fertility should be clearly stated right away.

Moreover, terms fitness/fertility appear sometimes alone, sometimes together (i.e., fitness (fecundity)) in the next sections and somehow that felt more confusing than helpful. Consistency may help.

Line 57 - following the previous suggestion 'a fitness component' can be removed.

Line 64 - would be great to see the hyp/predictions listed clearly (instead of point iv). The next paragraph would explain the rationale behind each hyp/prediction. This would help the reader follow the different tests/hyp later on.

Line 68 - this sentence repeats the idea in Line 51-53 correct?

Line 72 - S is missing in figure S2

Methods

Line 116 - how can you tell it is one colony per spot?

Line 139 to 144 - Maybe this paragraph should start subsection (e).

Plus overgrowth of colonies = interspecific competition? Again, for someone not very familiar with the system it is unclear what you mean right away.

Line 150 to 151 - odd sentence. Maybe change into something as 'The few female polymorphisms identified without ovicells were not accounted in the analyses (mean = (...))'.

Line 165, 166 - it reads (see electronic supplementary material) but no Suppl table/fig is indicated.

General comment Methods

From figure 2 legend and table S1 each time interval corresponds to a formation. Yet this is not clear in the text.

The time intervals (formations) have distinct spans. Also, the number of spots/samples is uneven. Overall, the authors do a good job of covering issues related to this. Same with dealing with environmental differences resulting from sample origin as well as preservation.

Results

Line 201 to 203 - how you obtain size or shape for the three traits is much clearer here than in the Methods section.

Moreover, size is used in some sections and area in the others. E.g., line 128 - 'Autozooid area we use for downstream analysis (...)'; line 201 - 'autozooid size (area approximated by length times width)'. Maybe use area (shape) across and when defining how you estimate this use a system closer to the Results format? Seems straight forward but on a first read, consistency helps...

Line 232 - crowding discussed but not in the title.

Discussion

Line 271 - where it reads 'more rapid' change to 'faster'.

Line 276 - missing punctuation after [7].

Line 294 - do you mean when fusion occurs new autozooids are narrower?

Line 308 to 310 - confusing sentence. Do you mean within-sample variance (colonies from substrates collected in the same location) is similar to variance resulting from multiple samples within a formation?

Line 313 to 319 - may be more appropriate in the Methods section. Plus cuts the Discussion flow a bit, which is a pity.

Figures

Fig 5 - legend indicates 4 estimates (word misspelt in legend) are significantly positive but 5 * are shown.

FigS10 - highlight Nukumaru Limestone formation so the note comment can be seen.

Author's Response to Decision Letter for (RSPB-2020-2047.R0)

See Appendix A.

RSPB-2020-2047.R1 (Revision)

Review form: Reviewer 1

Recommendation

Accept as is

Scientific importance: Is the manuscript an original and important contribution to its field?

Excellent

General interest: Is the paper of sufficient general interest?

Good

Quality of the paper: Is the overall quality of the paper suitable?

Excellent

Is the length of the paper justified?

Yes

Should the paper be seen by a specialist statistical reviewer?

No

Do you have any concerns about statistical analyses in this paper? If so, please specify them explicitly in your report.

No

It is a condition of publication that authors make their supporting data, code and materials available - either as supplementary material or hosted in an external repository. Please rate, if applicable, the supporting data on the following criteria.

Is it accessible?

Yes

Is it clear?

Yes

Is it adequate?

Yes

Do you have any ethical concerns with this paper?

No

Comments to the Author

The authors have successfully made revisions that improve the clarity of their message in order to address a broader audience. My remaining questions about the relative importance of within colony variation and data collected from smaller colony fragments does not impact of the interpretation of this study, though I do think that it points to an interesting question to be taken up in another independent study. That is a positive outcome.

Decision letter (RSPB-2020-2047.R1)

21-Dec-2020

Dear Dr Di Martino

I am pleased to inform you that your manuscript entitled "Trait-fitness associations do not predict within-species phenotypic evolution over 2 million years" has been accepted for publication in Proceedings B. Congratulations!!

Open Access

Paper charges

Sincerely,

Dr John Hutchinson

Associate Editor:

Board Member: 1

Comments to Author:

The referee is happy with your responses and revisions and believe that some concerns does not impact the findings in any significant way. Congratulations on your manuscript and Merry Christmas.

Appendix A

Answers to reviewers' comments

Associate Editor
Comments to Author:

Dear Authors

You will see that two reviewers have found the study of great interest and recommend publication. However, a number of major and minor comments are raised that could improve the study. In particular reviewer 1 raises some comments about intra-colony variability and the sampling of a colony, which seems necessary to explicitly clarify the nature of.

We thank associate editor for handling our manuscript and appreciate the constructive comments from the two reviewers. Below, we respond to the comments and state the changes we have made in our revised manuscript. Our responses are in blue. Note that the line numbers to which the reviewers refer correspond to the original (unrevised) manuscript, but not to the line numbers in the revised manuscript (but track-changes are visible such that edits are easy to see).

Reviewer(s)' Comments to Author:

Referee: 1

Comments to the Author(s)

This is a very impressive study. The research questions and context are important and the data and methods have the potential to address them. I believe that it could be an even stronger paper if a couple of key points were addressed, as well as some minor ones for clarity.

We thank R1 for these positive comments and for the constructive comments that follow.

Is colony wide variation adequately accounted for by measurements from a portion of the colony? In multiple separate studies, Cheetham, Hughes, Hageman and coauthors for each, have regularly documented that about 40-60% of variation for a morphometric character within a bryozoan species (or at least population/assemblage) accounted for by within colony (non-genetic) variation. These, combined and independent studies of O'Dea and Okamura and their co-authors, also document positional effects within colonies, both at the level of patches/spots within colonies and nearest neighbor effects of adjacent/nearby zooids. The authors of the present study need to comment on how emphasis of measurements concentrated at the spot level within a colony (~76% of colonies represented by measurement from a single ~5mm² "spot"), which represents a biased subset of the total within colony variation. Is one spot the size of a typical whole colony, or what proportion of a mature colony would a spot/fragment represent. These factors may not be relevant to conclusions of the paper, but the rationale for disregarding them, should be addressed logically and stated clearly.

Figures S6-9 attempt to constrain this, but the conclusions, implications taken from these figures needs to be better explained in the text. That is, do the authors believe that these results alleviate the concerns about measurements from a limited number of spots per colony adequately representing variation within colonies?

R1 makes good points that we should have been more explicit about. Yes, in e.g. Cheetham et al. 1995 Evolution, the authors, summarizing from several independent pieces of work, suggest that 50-70% of morphometric characters (e.g. linear measures of size that we have in this study) can be attributed to within-colony variation. The rest of the variation (30-50%) arise from inter-colony (interpreted as genetic) and environmental variation. Some of this 50-70% intracolony variation is attributed to “positional information”, where the first few zooids budding off from the first zooid (the ancestrula) are usually smaller than other later budded, “mature” zooids. However, the first few generations of zooids are not measured, so these do not contribute to the measured intracolony variation in our study. Note, however, that the variation in size of the zooids that bud under different environmental conditions in the same colony (notably temperature, see O’Dea and Okamura 1999, 2000 among others) is still due to intracolony variation (i.e. it is captured in Cheetham’s 50-70%).

The question we believe R1 is asking is whether measuring one to a few 5mm² spots for each colony (away from the zone of astogeny, i.e. away from the ancestrula) will capture the variation of **single** colonies and whether this will bias our results. We cordially argue that the first question is not quite the right question to ask in the context of our manuscript (as R1 also suggested by stating above that “these factors may not be relevant to conclusions”). What we would like to capture here is the relationship between measurements (e.g. autozooid size and ovicell size) not just within colonies, but **within** and **between** time intervals (separate panels in e.g. Figs. 2-5). This means that it is important to have measurements of many colonies, rather than a large part of single colonies. Random samples of spots within colonies are not biased towards specific parts of the colony (other than not representing early astogeny) and hence should not represent a biased sample with respect to the traits we are measuring.

That explained, R1 is right that we should explain our set up better. In the methods subsection now titled “(b) Measurement of average fecundity and phenotypic traits”, we clarify that the “spot” is a random sample within the given colony. In the caveats section of the Discussion, we add the text “We used one to several small random “spots” of preserved colonies to estimate morphological traits and fecundity (electronic supplementary material, tables S1 and S2), but note that much of the within-species variation is captured by only measuring few zooids within few colonies [24]. We also assume (see electronic supplementary material, figures S7–S9) that we have captured much of the 30–50% of trait variation that is external to within-colony variation [55].” Ref 24 is Liow and Taylor 2019 where data presented in the SI show that measurements from a single colony reflect much of the range of variation seen when many colonies (of the same species) from different time intervals are measured. Ref 55 is Cheetham et al. 1995 Evolution. We hope that these edits clarify our assumptions.

By the end the paper, “fitness” and ‘fit’ are treated as literal synonyms for ovicell size and number. I expect there is some truth in that conclusion, but the paper does not get far enough beyond the tautology to prove this. The logic that the number and or size of ovicells is a proxy for fecundity and that fecundity is related to (measure of?) fitness is elegant and simple. The literature on fitness both theoretic and measured is rich, but I adhere to the idea that the definition of fitness is an ever-moving evolutionary target. If larval recruitment between two taxa in a given setting, an ovicelled and non-ovicelled taxon, then in that circumstance, ovicell number is not a meaningful metric of relative fitness among species, so how does one demonstrate that it is within species?

Overgrowth “strategies” and fitness are more directly observed and evaluated, but even there the end game of “lesser fitness” has not resulted in the extinction of the cyclostomes. At the very least one could argue that some non-ovicellate taxa thrive in certain settings with alternative definitions of “fitness”.

The authors do not need to more fully demonstrate the biological fitness of a character trait, but I encourage them to be mindful that they are still working in a complex system and have not proven the overall fitness of an organism based on a single proxy. I am sure the author’s would agree, but again, I encourage them to avoid the tone that they have definitively defined fitness in this clade.

We thank R1 for urging us to be more conservative in our language (yes, we do agree we have not captured overall fitness) and we make edits to reflect the caution readers should take in interpretation.

We also agree with R1 that it is not possible to compare ovicellate and non-ovicellate taxa, which is not what we do here (we do not make this statement anywhere). Here we are comparing mean ovicell density among colonies of the same species. We **do** make the assumption, rooted on empirical results from previous studies, that ovicell density can be a meaningful metric of relative fitness among *colonies within the same species*. This we state in the introduction as “Here, we capitalize on the polymorphic, colonial nature of a metazoan group, namely cheilostome bryozoans, where we use the average density of female polymorphs within a genetic individual to estimate fecundity, which we use to approximate fitness [16]” where reference 16 is Yagunova & Ostrovsky 2010 Mar. Biol. Res. We tried to be clearer about our verbal usage of the term “fitness” at the beginning of our Discussion by inserting the text “In the following discussion, we use the term “fit” when interpreting results based on our estimated fecundity proxy, but note that this interpretation should be made cautiously (see also caveats section).”

We agree that the generalization concerning ovicells cannot be extended to species we have not studied and have stated throughout that it is only a component of fitness we have estimated (e.g. new lines 149, 268, 361). We add a sentence in the caveats section to address this, but also add that even studies based on extant populations do not (and mostly cannot) measure all possible components of fitness to get at true underlying fitness: we state “In addition, we acknowledge that we have only estimated one component of fitness as done in most studies based on natural populations (see examples in [8] Siepielski et al. 2009 Ecology Letters), namely fecundity, while life time fitness is also dependent on survival and growth.”

Other questions for clarification.

Line 105, Table S1- Clarify the methods used to calculate the temporal resolution. Probability statistic or just serial ranges?

It was a poor choice of wording on our part, apologies. With "temporal resolution" we meant the duration of the formation in million years, i.e. the time needed for the deposition of the formation which is given by the differences in ages between the lower boundary (max age of the formation) and the upper boundary (min age of the formation). We modified the text and changed "temporal resolution" to "temporal duration" (same wording we used in the Discussion) to avoid misunderstanding, and cited the reference (Naish et al 1998) where the estimates of the durations come from, in the main text.

Line 117 - Approximately how much area of a mature colony does a single “spot” represent? Recent, “undisturbed” colonies of *Antarctothoa tongima* growing on larger substrates are commonly circular and can (but seldom) reach a **maximum** diameter of 20 mm (Dennis P. Gordon, personal comment; also reported in the unpublished local guide of the Leigh Marine Laboratory, University of Auckland: Gordon, D.P. [1980] Bryozoa of the Cape Rodney to Okakari Point Marine Reserve: An identification manual. Unpublished Report, Wellington.) If we use the maximum diameter, then a spot represents about 1.5% of the maximum size of a colony (i.e. worst case scenario). While this does not seem like much, please remember that what we need is a random sample of “spots” in the colony **and** randomly sampled colonies to be able to answer our questions. One way to think about this is how many (or rather, few) people in any given population have their choice recorded in an exit poll, which can be an accurate reflection of the final tally of votes (and then again, we mustn’t base our inference on only one polling station). To be explicit about what is represented with the “spots”, we state in section (b) of the methods that “Note that maximum area to which this species grows is about 300 mm² (Dennis P. Gordon, pers. comm. 28.10.2020) such that the minimum area we have sampled is 1.6% (if we only sampled one spot for a given colony).”

Line 125 - Clarify that two spots off the same shell definitely represent two colonies (genotypes), i.e., these 311 groups are separate colonies (genotypes), not spots of which could be parts of a larger colony (shared genotype).

The text has been modified to state clearly that 311 is the number of distinct colonies, not spots (see also electronic supplementary material, table S2). These colonies were recognized as distinct via direct observations of separate ancestrulae (first zooids from which the colony starts growing) or opposite directions of growth when the ancestrula was missing or because placed on different surfaces of the shell (external vs internal). However, as we state in the caveats section (see Discussion): "We have also assumed that the colonies studied are genetically independent. Yet, if they are preserved on the same substrate or within the same sample, they may not be. However, it is known from microsatellite studies of the closely related *C. hyalina* that larvae settle randomly with respect to kin [53 = Hoare K, Hughes RN, Goldson AJ. 1999 Molecular genetic evidence for the prevalence of outcrossing in the hermaphroditic brooding bryozoan *Celleporella hyalina*. Mar. Ecol. Prog. Ser. 188, 73–79. (doi:10.3354/meps188073)]"

Line 268 - It would be useful to see a calculation of water volume clearance for small vs. large lophophores, both at closest packing. Water volume clearance per time is the key, not just area.

We agree with R1 that it would be interesting to know the difference in water volume clearance between large vs small lophophores. However, our data (i.e. counts of polymorphs and size of autozooids and ovicells) do not give us any insights regarding the sizes of the lophophore. The size of the orifice (the opening through which the polypide can extrude to feed) is thought to be a good proxy for lophophore size, which is in turn a good proxy for the size of suspended food particles the polypide can process (see Jackson & McKinney 1989). There is no demonstrated quantitative relationship between autozooid size and lophophore size. But what we know (and what we meant to communicate) is that smaller zooids correspond to more numerous individual zooids occupying the same space, which in turn means more polypides/lophophores per surface unit (see also [43 = Okamura B. 1987 Seasonal changes in zooid size and feeding activity in epifaunal communities of *Electra pilosa*. In International Bryozoan Conference (ed JPR Ross), pp. 197–203. Bellington, Washington: Western Washington University].

Line 268 - reword, “why do colonies with more & larger ovicells have smaller autozooids?” We are unsure why R1 wants us to reword this. (We believe the sentence in the original line 268 is “Why are colonies with smaller autozooids fitter” which R1 suggests replacing with the given sentence. Please let us know if we are mistaken. We revised this sentence as “Why are colonies with smaller autozooids fitter, in the sense that they can have more offspring at any one time, judging by the density of ovicells?” for clarity instead.

Line 295 - is this just canalization from packing constraints, or some other controlling factor? R1 seems to be referring to the sentence “Here, we might speculate that skinnier autozooids are a reflection of higher growth rates especially when meeting conspecifics to increase the chance of contact and fusion between colonies” given the line number cited. We understand canalization as the tendency for development to follow particular (the same) trajectory despite external or internal perturbation.” (e.g. Siegal and Bergman 2002 PNAS). Given that there is a change in phenotype, we are not quite sure what R1 is referring to by canalization. We suggested (based on reference 49) that triggers/cues from conspecifics result in high growth rates that result in skinner zooids.

Line 275 - are the authors calling on ambient temperature as the control (environmental) of the autozooid size phenotype? or calling on evolutionary changes to the genotype as a fitness response?

The text has been modified to clarify this point. We now specify that $\delta^{18}\text{O}$ approximates ambient relative sea water temperature thus providing a good record of the climate which explains part of the temporal autozooid size variation.

Line 276 - Do not use the implied content of citation “[7]” as the subject in the sentence. A full stop was missing at the end of this sentence, apologies. This probably created some confusion. Citation [7] is not the subject of the sentence, we instead compare our autozooid size-temperature results with similar results in Hunt et al. 2015, based on a different system (ostracodes).

Line 128 – What is the “downstream” analysis?

The text has been modified for clarity. What we meant was “subsequent” analyses.

Line 134 – Please clarify, is “highly accurate” measurement replication a function of single researcher’s replicates or replication among multiple workers measuring the same material? (typically a big difference).

We refer to a single researcher on separate occasions and we modified the text to reflect this. Consistency to reflect relative difference is important in our context hence only one person made the measurements, which are demonstrated to be self-consistent. Hence, although we do not have estimates of the confidence of the absolute measurements, we have high confidence in the consistency of the repeatability of relative measurements (which is what is important here).

Referee: 2

Comments to the Author(s)

The present work analyses the correlation between fitness (measured as fecundity) and other traits in the deep time. The study system is excellent since few taxa have good enough fossil material for the proposed aim, added to work on living relatives that support the

hyp/assumptions presented. The study is of interest to the broad readership of Proceedings B.

I recommend publication and have only minor comments/suggestions before the paper can be published.

We thank R2 for these positive comments and for the constructive suggestions that follow.

Abstract

Line 24 - paleoclimate is too general to be informative in this context, I think. Maybe clarify which proxy used to reconstruct past climate variation. Ecological interaction is a broad term, interspecific competition or overgrowth seems clearer.

We modified the abstract to specify that palaeoclimate is approximated using $\delta^{18}\text{O}$ and also specified overgrowth competition and substrate crowding in place of the generic “ecological interactions”. We modified throughout the text accordingly in order to be more explicit and consistent with respect to this comment.

Background

Line 51 - component of fitness measured as average female fertility should be clearly stated right away.

Good idea, we state this in the background section of the introduction as “Using an exceptional study system, we assess, for the first time, the association between a fitness component (fecundity) and phenotypic traits for fossil populations...” Then immediately, the following sentence, we expand on this as “Here, we capitalize on the polymorphic, colonial nature of a metazoan group, namely cheilostome bryozoans, where we use the average density of female polymorphs within a genetic individual to estimate fecundity, which we use to approximate a component of fitness.”

Moreover, terms fitness/fertility appear sometimes alone, sometimes together (i.e., fitness (fecundity)) in the next sections and somehow that felt more confusing than helpful. Consistency may help.

We do have to use the term “trait-fitness” association and cannot substitute it with “trait-fecundity” associations in this context, but we tried our best throughout to minimize confusion and to be consistent throughout. Essentially, the fitness is a fitness component, which in our case is estimated by fecundity, which is in turn estimated by average ovicell density. This “chain” is a bit too cumbersome to repeat (and we are sure R2 sees the logic in this) once we have explained this in both the introduction (see changes associated with the comment above) and the methods. Also, note that we never used the term “fertility” but fecundity (a commonly measured fitness component among contemporary populations of diverse species).

Line 57 - following the previous suggestion, a fitness component' can be removed. Removed and text modified.

Line 64 - would be great to see the hyp/predictions listed clearly (instead of point iv). The next paragraph would explain the rationale behind each hyp/prediction. This would help the reader follow the different tests/hyp later on.

Thanks for the great suggestion. We modified the organization of the text (last paragraph of Background) accordingly. We removed point (iv), and the following redundant statement (see comment below), to make it easier for the reader to follow the rationale concerning our hypotheses and predictions.

Line 68 - this sentence repeats the idea in Line 51-53 correct?

We agree with R2 that this sentence is redundant and thus we removed it.

Line 72 - S is missing in figure S2

Added.

Methods

Line 116 - how can you tell it is one colony per spot?

A typical example of what we call a "spot" is shown in electronic supplementary figure S1. As you can see from figure S1, the zooids are growing continuously, there are not discrete groups of zooids (clusters of zooids totally separated from each other), and there is no trace of different growth directions. These (geometrical) observations ensure that we are looking at a single colony for each spot.

Line 139 to 144 - Maybe this paragraph should start subsection (e).

As suggested by R2 we moved this paragraph from subsection (b) to subsection (e) and re-arranged the text.

Plus overgrowth of colonies = interspecific competition? Again, for someone not very familiar with the system it is unclear what you mean right away.

The text has been modified (see comment above). But note also that we introduced overgrowth interactions in the abstract and background but also more consistently throughout the text for clarity.

Line 150 to 151 - odd sentence. Maybe change into something as 'The few female polymorphisms identified without ovicells were not accounted in the analyses (mean = (...))'. Text changed accordingly. Now it reads "Note that the few female polymorphs not bearing ovicells (mean = 5.9%, median = 0 % of all measured colonies with females observed) were not included in the analyses, as they have not contributed to offspring by the time of death"

Line 165, 166 - it reads (see electronic supplementary material) but no Suppl table/fig is indicated.

References to the specific suppl. figures/tables were added.

General comment Methods

From figure 2 legend and table S1 each time interval corresponds to a formation. Yet this is not clear in the text.

Text in Methods, section (a), second paragraph was modified to clarify this point. Now it reads "We studied 414 fossil colonies of *A. tongima* collected in January 2014 and March 2017 from six Pleistocene formations of the Wanganui Basin, corresponding to six time intervals dated from 2.29 to 0.30 million years ago (Ma) [32], with a temporal duration ranging from 0.11 to 0.02 million years (myr) (electronic supplementary material, table S1)."

The time intervals (formations) have distinct spans. Also, the number of spots/samples is uneven. Overall, the authors do a good job of covering issues related to this. Same with dealing with environmental differences resulting from sample origin as well as preservation.

Thank you!

Results

Line 201 to 203 - how you obtain size or shape for the three traits is much clearer here than in the Methods section.

We agree with R2 and we modified/simplified the text in the Methods section accordingly. Now it reads "Autozooid and ovicell sizes (i.e. areas) were estimated by multiplying maximum length by maximum width. For ovicell size...[...] Autozooid shape was obtained as maximum length divided by maximum width."

Moreover, size is used in some sections and area in the others. E.g., line 128 - 'Autozooid area we use for downstream analysis (...)' ; line 201 - 'autozooid size (area approximated by length times width)'. Maybe use area (shape) across and when defining how you estimate this use a system closer to the Results format? Seems straight forward but on a first read, consistency helps...

We have modified text throughout for consistency as revision for other comments. Just to reiterate: Area = L times W, Shape = L/W, which we state early on.

Line 232 - crowding discussed but not in the title.

We have modified the subtitle of Results section (c) to reflect this. Now it reads " **$\delta^{18}\text{O}$, but not overgrowth competition and crowding, effects phenotypic traits and fecundity**"

Discussion

Line 271 - where it reads 'more rapid' change to 'faster'.

Text changed.

Line 276 - missing punctuation after [7].

Added.

Line 294 - do you mean when fusion occurs new autozooids are narrower?

In this paragraph of the Discussion we speculate on the reason behind the positive relationship between intraspecific interactions (the presence of other colonies of *A. tongima* on the same substrate) and skinnier/longer autozooids. We refer to the fact that allorecognition has been invoked for bryozoans and other marine invertebrates, which will allow colonies (under certain genetic requirements) to recognize the presence of another colony of the same species in order to facilitate the fusion of the two colonies together. This topic is elaborated in Manriquez PH. 1999 Mate choice and reproductive investment in the cheilostome bryozoan *Celleporella hyalina* (L.). University of Wales, Bangor, Gwynedd. Available here <http://e.bangor.ac.uk/id/eprint/4130>.

Line 308 to 310 - confusing sentence. Do you mean within-sample variance (colonies from substrates collected in the same location) is similar to variance resulting from multiple samples within a formation?

We agree with R2 that the sentence was not clear. Text has been modified according to this suggestion (and previous) suggestions. Now it reads "However, within-sample trait variances (representing a relatively short amount of time) are not distinguishable from those resulting from multiple samples (representing a greater amount of time) within a formation (electronic supplementary material, table S3)."

Line 313 to 319 - may be more appropriate in the Methods section. Plus cuts the Discussion flow a bit, which is a pity.

We would prefer to leave this paragraph here in the Discussion section because we believe it is a part of the caveats more than the Methods.

Figures

Fig 5 - legend indicates 4 estimates (word misspelt in legend) are significantly positive but 5 * are shown.

Indeed, 5 out of 6 estimates are significant for autozoid shape but only 4 are significantly positive. Typos in the legend have been corrected.

FigS10 - highlight Nukumaru Limestone formation so the note comment can be seen.

Red circles have been added in this figure to highlight the mentioned formation.